# Non-Coding RNAs of Extracellular Vesicles: Key Players in Organ-Specific Metastasis and Clinical Implications

**DOI:** 10.3390/cancers14225693

**Published:** 2022-11-19

**Authors:** Qian Jiang, Xiao-Ping Tan, Cai-Hua Zhang, Zhi-Yuan Li, Du Li, Yan Xu, Yu Xuan Liu, Lingzhi Wang, Zhaowu Ma

**Affiliations:** 1Department of Gastroenterology, First Affiliated Hospital of Yangtze University, Health Science Center, Yangtze University, Jingzhou 434023, China; 2Digestive Disease Research Institution of Yangtze University, Yangtze University, Jingzhou 434023, China; 3Department of Cardiovascular Medicine, Honghu Hospital of Traditional Chinese Medicine, Honghu 433200, China; 4School of Basic Medicine, Health Science Center, Yangtze University, Jingzhou 434023, China; 5Department of Pharmacology, Yong Loo Lin School of Medicine, National University of Singapore, Singapore 117600, Singapore; 6Cancer Science Institute of Singapore, National University of Singapore, Singapore 117599, Singapore; 7NUS Centre for Cancer Research (N2CR), National University of Singapore, Singapore 117599, Singapore

**Keywords:** extracellular vesicles, noncoding RNAs, organ-specific metastasis, cancer biomarkers, anti-cancer metastasis

## Abstract

**Simple Summary:**

Metastasis refers to the progressive dissemination of primary tumor cells and their colonization of other tissues, and is a major cause of treatment failure in cancer patients. Most cancers metastasize to specific organs with a non-random distribution pattern through a process known as “organ-specific metastasis”. Extracellular vesicles (EVs), mediating intercellular communication, can deliver ncRNAs to regulate cancer phenotypes at distant organs or sites in multiple cancers, thereby contributing to organ-specific metastasis. This review summarizes the underlying mechanisms and functions of EV-ncRNA-mediated metastatic organotropism in bone, liver, lung, brain, and lymphatic metastasis, and highlights the clinical applications of EV-ncRNAs serving as potential biomarkers and therapeutic targets. The aim of this review is to provide a perspective on the novel therapeutic strategies of organ-specific metastasis and hope for improving the survival outcome of cancer patients.

**Abstract:**

Extracellular vesicles (EVs) are heterogeneous membrane-encapsulated vesicles released by most cells. They act as multifunctional regulators of intercellular communication by delivering bioactive molecules, including non-coding RNAs (ncRNAs). Metastasis is a major cause of cancer-related death. Most cancer cells disseminate and colonize a specific target organ via EVs, a process known as “organ-specific metastasis”. Mounting evidence has shown that EVs are enriched with ncRNAs, and various EV-ncRNAs derived from tumor cells influence organ-specific metastasis via different mechanisms. Due to the tissue-specific expression of EV-ncRNAs, they could be used as potential biomarkers and therapeutic targets for the treatment of tumor metastasis in various types of cancer. In this review, we have discussed the underlying mechanisms of EV-delivered ncRNAs in the most common organ-specific metastases of liver, bone, lung, brain, and lymph nodes. Moreover, we summarize the potential clinical applications of EV-ncRNAs in organ-specific metastasis to fill the gap between benches and bedsides.

## 1. Introduction

Extracellular vesicles (EVs) are membrane vesicles secreted by almost all types of cells. Based on the size and biogenesis pathway, EVs can be classified into apoptotic bodies, microvesicles, and exosomes [1,2]. Under distinct physiological and pathological conditions, EVs act as key mediators of intercellular communication by delivering various cargoes, including nucleic acids, lipids, proteins, and other bioactive molecules between cells [3,4,5]. Mounting evidence has shown that diverse molecules delivered by EVs can contribute to metastasis and organotropism, which alters the tumor microenvironment (TME) and cancer progression [6,7]. Non-coding RNAs (ncRNAs) such as microRNAs (miRNAs), long noncoding RNAs (lncRNAs), and circular RNAs (circRNAs) play diverse and context-dependent roles in cancer [8]. NcRNAs can regulate cancer progression outside the primary cancer cells via EV-mediated transfer to recipient cells [9]. EV-ncRNAs are master regulators of various cellular processes and play an important role during carcinogenesis [10]. Tumor cell-derived EV-ncRNAs influence epithelial-mesenchymal transition (EMT), cell proliferation, metastasis, angiogenesis, drug resistance, and inflammation [11].

Metastasis is the leading cause of cancer-related death and is the most catastrophic feature of cancer [12,13,14]. Metastasis refers to the progressive dissemination of tumor cells from the primary tumor site and the colonization of other organs or tissues by the tumor cells [15]. Organ-specific metastasis refers to the ability of the primary tumors to induce and control secondary tumors at the metastatic site in a specific organ [16], which involves a series of related invasion and metastasis events [17]. Studies have shown that tumor-derived exosomes can promote organ-specific metastasis [18]. Various studies have shown that EV-derived ncRNAs can regulate cancer phenotypes at distant organs or sites in multiple cancers, thereby contributing to organ-specific metastasis.

Recently, a few reviews have focused on the general role of EVs in organ-specific metastasis [6,16,19]; however, the underlying mechanisms and role of EV-derived ncRNAs have not been comprehensively explored in organ-specific metastasis. In this review, we summarized the functions of EV-derived ncRNAs in organ-specific metastasis and highlighted their potential applications in cancer diagnosis and therapeutics.

## 2. EVs, ncRNAs, and Organ-Specific Metastasis

### 2.1. Characteristics of EVs and ncRNAs

EVs are lipid bilayer membrane vesicles secreted by almost all cell types, including tumor cells [20,21]. In the past decade, EVs have been classified into three major categories based on their size, biogenesis, and release pathways: (i) exosomes with a diameter of 30–150 nm, (ii) microvesicles with a diameter of 0.1~1.0 μm, (iii) apoptotic bodies with a diameter of 1–4 μm [3,22]. Exosomes are the most extensively studied subtype of EVs and are vesicles formed by the fusion of multivesicular bodies and plasma membranes. Microvesicles are generated by the outward budding of plasma membranes [23]. Among several factors associated with the mechanism of exosomal biogenesis, the endosomal sorting complex required for transport (ESCRT) is the most recognized regulator of exosomal biogenesis [24]. EVs can alter cell fate by delivering proteins, nucleic acids, transcription factors, and ncRNAs to recipient cells for intercellular communication via autocrine, paracrine, and endocrine signals [25]. EVs are involved in the various processes associated with cancer progression, including inflammatory response, immune suppression, angiogenesis and lymphogenesis, cell migration and proliferation, invasion, EMT, and metastasis [26]. Recent studies have indicated that EV-ncRNAs mediate tumor growth and metastasis by establishing communication between tumor cells and the TME.

NcRNAs lack protein-coding potential and primarily include miRNAs, lncRNAs, circRNAs, PIWI-interacting RNAs (piRNAs), and transfer RNA (tRNA)-derived small RNAs (tsRNAs) [8,11]. NcRNAs can influence cancer progression by regulating gene expression at epigenetic, transcriptional, and post-transcriptional levels [8]. MiRNAs are approximately 22 nucleotide long endogenous small ncRNAs [27], which are loaded onto a protein complex called the RNA-Induced Silencing Complex (RISC) by complementary binding to the 3’-untranslated regions of the target mRNAs. MiRNAs regulate target gene expressions post-transcriptionally by triggering mRNA degradation or inhibiting protein translation [28,29]. LncRNAs are approximately 200-nucleotide-long transcripts. They are involved in several processes such as chromosomal silencing, transcriptional regulation, chromatin modification, and nuclear transport by interacting with DNA, RNA, and proteins [30]. LncRNAs have been identified as key biologically active molecules in exosomes derived from tumor cells [31]. Exosomal lncRNAs can modify the TME and play a central role in tumor progression by regulating tumor growth, metastasis, and angiogenesis [32]. CircRNAs are non-traditional ncRNAs that generated from post-splicing of the primary transcript with a covalent closed-loop structure. CircRNAs can regulate transcription and splicing, modulate cytoplasmic mRNA stability, sequester miRNAs, and serve as templates for translation in diverse biological and pathophysiological processes [33,34,35]. NcRNAs can be enclosed in EVs and are highly stable, hence, can be promising diagnostic biomarkers. Studies have shown that the ncRNA networks can influence multiple molecular targets and have been identified as carcinogens and tumor suppressors in various cancers [36]. Studies have demonstrated that ncRNAs play an important role at all stages during tumor metastasis and could serve as key therapeutic targets for many cancers [37].

### 2.2. EV-Derived ncRNAs: Key Players in Organ-Specific Metastasis

Cancer metastasis involves several steps. Metastasis begins with tumor cells invading local tissues surrounding the primary tumor site. The next step is intravasation, where the cancer cells enter the blood circulation via the lymphatic system or blood vessels. To survive and disseminate to different organs, the tumor cells exit the circulation via a process called extravasation into the parenchyma of distant organs. Once the cancer cell has extravasated, they either die, remain dormant, or proliferate to colonize distant organs [38]. Metastasis is an important feature of malignant cancers and the leading cause of cancer-related death. Metastasis follows a non-random distribution pattern among distant organs, called “organotropic metastasis”, “metastatic organotropism”, or “organ-specific metastasis”. Different cancers and their subtypes show distinct organotropy [39]. In the 1990s, Paget’s “seed and soil” hypothesis established a conceptual framework for organ-specific metastasis [40]. Later, it was proposed that cancer cell seeds are intrinsically compatible with specific tissues’ microenvironmental soils, which aids in determining metastatic organotropism [40,41]. In 2005, Dr. Lyden and colleagues proposed the term “pre-metastatic niche (PMN)” to describe the phenomenon that primary tumors metastasize by recruiting bone marrow-derived cells (BMDCs) to distant organs and establishing a microenvironment conducive to metastasis [42].

Primary tumor cells secrete EVs as messengers into circulation prior to the initiation of metastasis [43]. Subsequently, primary tumor cells disseminate and colonize specific target organs via EVs, and fuse with target cells to induce organ-specific metastasis [44]. The EV-mediated transmission pathways can drive metastasis in bone, liver, lung, brain, and lymph nodes, which are the most common sites for cancer organ-specific metastasis (Figure 1). The organ-specific metastasis involves interactions between cancer cells and the host microenvironment, including the activation of the paracrine cytokine loop, modification of host cell composition, and altering the extracellular matrix structure [45]. Some studies suggest EVs are biased towards specific organs due to their adhesion by fusing with specific resident cells, thereby preparing favorable PMN for further metastasis, such as exosomal integrin αvβ5 determining liver metastasis [18]. Another studies have exhibited that prostate cancer (PCa) have an exceptionally high chance of developing bone metastasis [46], while breast cancer (BC) can metastasize to the lung, liver, bone, or brain [45,47]. Likewise, brain metastases are common in patients with lung and breast cancers [48]. Hoshino et al. proposed that tumor-derived exosomes affect organ-specific metastasis by two mechanisms: the intrinsic genetic makeup of cells and the PMN [49]. Tumor-derived exosomes modulate local and systemic TME and mediate organ-specific metastasis by transferring specific bioactive molecules, including ncRNAs [50].

Recent studies suggest that EV-derived ncRNAs play a crucial role in the organ-specific metastasis of many cancers. Exosomes derived from Fusobacterium nucleatum-infected colorectal cancer (CRC) cells containing miR-1246/92b-3p/27a-3p and CXCL16 promote tumor growth and liver metastasis of CRC [51]. Further, castration-resistant PCa cell-derived exosomal HOXD-AS1 promotes bone metastasis in PCa by modulating the miR-361-5p/FOXM1 axis [52]. The exosomal circNRIP1 sponges miR-149-5p to promote lung metastasis in gastric cancer (GC) via the AKT1/mTOR signaling pathway in patient-derived xenograft mouse models [53]. Several studies have shown that EV-derived ncRNAs can regulate organ-specific metastasis by altering various oncogenes and tumor suppressors. Herein, we summarized the underlying mechanisms and functions of EV-derived ncRNAs in organ-specific metastasis (Table 1), including bone, liver, lung, brain, and lymphatic metastases.

## 3. Emerging Functions of EV-Derived ncRNAs in Organ-Specific Metastasis

### 3.1. Bone Metastasis

Bone is the most common site for the metastasis of various cancers, including PCa, bladder cancers (BCa), multiple myeloma (MM), and lung cancers [113]. Bone metastasis is a multistage process, which involves the colonization of the bone marrow by tumor cells, adapting to the local microenvironment, the establishment of cancer niche, and the disruption of normal bone homeostasis via interplay between tumor cells and various bone cells such as osteoclasts, osteoblasts, and osteocytes [114]. In the subsequent sections, we have summarized the underlying mechanisms and emerging functions of EV-derived ncRNAs in bone metastasis (Figure 2A and Table 1).

#### 3.1.1. Bone Metastasis of Prostate Cancers

Bone metastasis commonly occurs in patients with PCa, and is one of the devastating complications of PCa [115]. Various studies have shown the functions and mechanisms of exosomal ncRNAs in the bone metastasis of PCa. A study has shown that PCa cell-derived miR-378-3p containing EVs promotes osteolytic progression in the bone metastasis of PCa by activating the DYRK1A/NFATC1/ANGPTL2 axis in the macrophages of bone marrow [54]. Ma et al. have shown that PCa-derived small EVs (sEVs) containing miR152-3p transmit osteolytic signals to osteoclasts and promote osteolytic progression in bone metastasis by directly targeting the osteoclastogenesis regulator MAFB [55]. Further, Yu et al. have shown that PCa-derived exosomes carrying miR-92a-1-5p promote bone metastasis by enhancing osteoclast differentiation and suppressing osteoblastogenesis by downregulating type I collagen expression [56]. PCa-derived exosomal miR-375 mediates bone metastasis in patients with PCa by increasing osteoblast activity [57]. Tumor-derived EVs can alter the pre-metastatic bone microenvironment by inducing phenotype-specific differentiation [6]. Hashimoto et al. performed a series of functional assays and showed that EVs containing miR-940 derived from PCa cells induce osteoblast phenotype differentiation and promote bone metastasis by targeting ARHGAP1 and FAM134A in human plasmic stem cells [58]. Ye et al. showed that miR-141-3p enriched in the exosomes derived from MDAPCa 2b cell promotes osteoblast metastasis by regulating osteoblast activity. Mechanistically, miR-141-3p regulates the bone metastasis microenvironment by inhibiting DLC1 expression. This leads to the activation of the p38MAPK signaling pathway and increases OPG/RANKL expression, thereby directly enhancing the osteoblast’s activity and indirectly inhibiting osteoclasts’ activity [59]. A high miR-210-3p expression was observed in the bone metastasis tissues of PCa. miR-210-3p activates the NF-κB signaling pathway by targeting its negative regulators such as TNIP1 and SOCS1, thereby inducing the bone metastasis of PCa cells [60]. PCa cell-derived EVs containing lncRNA can modify the bone microenvironment and promote the bone metastases of PCa cells [116,117]. A study by Mo et al. shows that PCa-derived exosomes containing NEAT1 can promote the osteogenic differentiation of human bone marrow-derived mesenchymal stem cells via miR-205-5p/RUNX2 and SFPQ/PTBP2 axis [61]. Hu et al. revealed that lncRNA NORAD was highly expressed by the EVs derived from PCa cells, which promotes the bone metastasis of PCa cells via the miR-541-3p/PKM2 axis [62]. Furthermore, castration-resistant PCa cell-derived exosomal HOXD-AS1 promotes the bone metastasis of PCa [52].

#### 3.1.2. Bone Metastasis of Breast Cancer

BC was the most commonly diagnosed cancer in 2020. Further, the incidence of BCs was the highest [118,119]. Bone is the most common site for the distant metastasis of BC [113], thus, increasing mortality in patients with BC [114]. Recent studies have shown that BC-derived exosomal miRNAs play a key role in regulating bone metastasis in patients with BC. A high expression of exosomal miR-21 was observed in the serum of patients with bone metastasis of BC. Exosomal miR-21 regulates the generation of osteoclasts by directly targeting programmed cell death 4 (PDCD4), which establishes a PMN to promote bone metastasis [63]. Among different subtypes of BC, bone metastasis commonly occurs in patients with estrogen receptor-positive (ER+) BC [120]. A study has shown a significant positive correlation between the high expression of exosomal miR-19a and bone metastasis in ER^+^ BC cells. IBSP recruits osteoclast precursors to form a metastatic niche enriched in osteoclast precursors. IBSP transports exosomal miR-19a to osteoclast precursors and promotes the activation of the NF-κB and AKT signaling pathways by inhibiting PTEN expression. This induces osteoclast generation and creates a microenvironment favorable for bone metastasis [64]. Further, the high expression of exosomal miR-20A-5p derived from BC cells was observed in both tumor tissues and MDA-MB-231 cells, which promotes osteoclast generation and bone metastasis by targeting SRCIN1 [65].

#### 3.1.3. Bone Metastasis of Lung Cancer

In 2020, lung cancer was the second most commonly diagnosed cancer and the leading cause of cancer-related death worldwide [118]. Bone is a common site for the metastasis of lung cancer, which significantly affects the survival and quality of life of the patients [113]. Non-small cell lung cancer (NSCLC) accounts for 85% of lung cancer cases [121]. Approximately 40% of NSCLCs develop bone metastasis [122]. A significant increase in NSCLC cell-derived exosomal miR-17-5p was observed in NSCLC cells and tissues, which promotes osteoclastogenesis. Mechanistically, miR-17-5p directly targets PTEN and activates the PI3K/AKT signaling pathway to enhance osteoclast differentiation and promote bone metastasis in NSCLC [66]. Exosomal miR-21 derived from lung cancer cells targets PDCD4 to promote osteoclastogenesis and the bone metastasis of lung cancer [67]. A study has reported a high expression of lncRNA-SOX2OT in exosomes derived from NSCLC patients with bone metastasis. Exosomal lncRNA-SOX2OT induces osteoclast differentiation by targeting the miR-194-5p/RAC1 axis and the tumor growth factor-β1 (TGF-β1)/pTHrP/RANKL signaling pathway, thereby promoting bone metastasis in patients with NSCLC [68].

#### 3.1.4. Bone Metastasis of Other Cancer

Bone marrow is a common site for neuroblastoma metastasis, and bone marrow metastasis is associated with poor outcomes in patients with neuroblastoma [123]. Marta et al. showed that high miR-375 expression downregulates YAP1 levels and induces the osteogenic differentiation of mesenchymal stromal cells to promote bone metastasis [69]. MM is a clonal plasma cell malignancy associated with osteolytic bone disease. Recent studies have demonstrated the role of MM-derived exosomes in osteoclastogenesis. The overexpression of miR-21 in bone marrow-derived mesenchymal stem cells adherent to MM cells contributes to bone metastasis by enhancing the activation of the STAT3-mediated receptor activator of NF-κB ligand RANKL and mediating RANKL-induced osteoclastogenesis [70].

### 3.2. Liver Metastasis

The liver is one of the most common sites for the metastasis of various cancers, including CRC, GC, and lung cancer [124]. The hepatic microenvironment provides autocrine and paracrine signals originating from both parenchymal and nonparenchymal cells that create a PMN for developing liver metastases [125]. The progression of liver metastasis involves two steps: creating a PMN and tumor invasion [126]. Studies have shown that tumor-derived EVs participate in the establishment of a PMN in the liver [75]. EVs regulate liver metastasis by activating the pro-inflammatory pathways and recruiting stromal cells [16,78]. In the following sections, we have summarized the role of EV-derived ncRNAs as multi-functional regulators transported to the liver from different organs by EVs, which can manipulate liver metastasis (Figure 2B and Table 1).

#### 3.2.1. Liver Metastasis of Colorectal Cancer

The liver is the most common site for metastasis of CRC [124]. The liver metastasis of CRC (CRLM) is the most common cause of CRC-related mortality and accounts for half of CRC-related deaths [127]. CRLM typically occurs due to interactions between CRC cells and the TME of the liver. However, the molecular mechanisms underlying the crosstalk between tumor-derived EV-containing miRNAs and the TME in CRLM are still unclear. Zhao et al. have shown a positive correlation between high levels of EV containing miR-181a-5p and liver metastasis in patients with CRC. Exosomal miR-181a-5p activates hepatic stellate cells by targeting SOCS3, which induce the remodeling of TME and promote liver metastasis [71]. Additionally, a significant increase in exosomal miR-934 levels was observed in patients with CRC and CRLM. Further, CRC-derived exosomal miR-934 directly downregulates PTEN expression and activates the PI3K/AKT signaling pathway, which induces the polarization of M2 macrophages. This leads to the activation of CXCL13/CXCR5/NF-κB p65/miR-934 positive feedback loop, thereby promoting CRLM [72]. A high expression of exosomes carrying miR-221/222 was observed in patients with CRC and liver metastasis. CRC-derived exosomes carrying miR-221/222 activate hepatocyte growth factor (HGF) by suppressing SPINT1 expression, which promotes the formation of a PMN and accelerates the progression of CRLM [73]. Liu et al. reported a low expression of exosomal miR-140-3p in the plasma of patients with CRC. Exosomal miR-140-3p targets B-cell lymphoma-2 (BCL2) and BCL9 to inhibit CRC progression and liver metastasis [74]. Further, CRC-EV-miR-21-5p promotes CRLM by activating the miR-21/TLR7/IL-6 axis to induce the formation of a pro-inflammatory PMN [75]. Zeng et al. showed that CRC-derived exosomal miR-25-3p regulates VEGFR2, tight junction protein zonula occludens-1 (ZO-1), OCCLUDIN, and CLAUDIN 5 expression in endothelial cells by downregulating KLF2 and KLF4 expression. This increases vascular permeability and angiogenesis and induces pre-metastatic ecotone formation, thereby promoting liver metastasis in CRC [76].

#### 3.2.2. Liver Metastasis of Gastric Cancer

GCs are the fourth leading cause of cancer-related mortalities worldwide [118]. Approximately 4–11% of patients with gastric cancer are detected with liver metastases at initial diagnosis [128]. Liver metastasis severely affects the prognosis of patients with GC. Specific sEV-miRNAs secreted by several cancers induce the formation of PMN in target organs before tumor cell colonization. Li et al. have reported the overexpression of GC cell-derived sEV-miR-151a-3p in the plasma of patients with liver metastasis, which can promote the condition. Mechanistically, sEV-miR-151a-3p inhibits SP3 activity by targeting YTHDF3, which activates the SMAD2/3 signaling pathway and enhances the stem cell-like properties of afferent GC cells. This aids in establishing a niche for accelerating liver metastasis in patients with GC [77]. Zhang et al. have elucidated a novel mechanism of liver metastasis in GC, wherein the exosomes containing epidermal growth factor receptor (EGFR) secreted by GC cells can be transported to the liver and integrated into the plasma membrane of stromal cells. The translocation of EGFR regulates the liver microenvironment and promotes liver metastasis by downregulating miR-26a/b expression and activating HGF [78].

#### 3.2.3. Liver Metastasis of Other Cancers

Liver metastasis occurs in 20–30% of patients with NSCLC, and the overall survival of patients is less compared to other metastatic sites [129]. Further, liver metastasis indicates poor progression-free survival in NSCLC patients treated with first-line cytotoxic chemotherapy [130]. Jiang et al. demonstrated a significant increase in lncRNA-ALAHM levels in EV derived from lung adenocarcinoma (LUAD) cells. EV overexpressing lncRNA-ALAHM binds to AUF1 to promote HGF secretion by hepatocytes, thereby facilitating the liver metastasis of LUAD cells [79]. A high expression of exosomal circ-IARS was observed in patients with pancreatic cancer and liver metastasis. Exosomal circ-IARS sponges miR-122, thereby decreasing the expression of ZO-1, increasing Rho A activity and F-actin expression, thereby altering the permeability of endothelial monolayers to promote liver metastasis [80].

### 3.3. Lung Metastasis

The lungs are the common site for the metastasis of many cancers, including BC, osteosarcoma, and gastrointestinal tumors [45]. EVs derived from metastatic cancer cells promote lung metastasis primarily by recruiting immune cells, vascular remodeling, and stromal alteration [16]. In the next section, we have summarized the underlying mechanism and role of EV-derived ncRNAs in lung metastasis (Figure 1C and Table 1).

#### 3.3.1. Lung Metastasis of Breast Cancer

Lung metastasis is the leading cause of death in patients with BC [131]. Recent studies have shown that EV-derived ncRNAs play a role in the metastasis of BC to the lungs. Zhou et al. show that exosomal miR-105 derived from metastatic BC regulates the migration of endothelial cells. Exosomal miR-105 enhances vascular permeability and promotes lung metastasis by downregulating tight junction protein ZO-1 and disrupting the barrier created by the vascular endothelial monolayer [81]. Vu et al. showed that a high expression of miR-125b in EVs derived from BC cells activates cancer-associated fibroblasts by inhibiting p53 and TP53INP1 expression, thereby promoting lung metastasis [82]. Gu et al. demonstrated that BC cell-derived exosomes containing miR-200b-3p activate the AKT/NF-κB p65 signaling pathway by inhibiting PTEN expression, thereby promoting the expression of CC motif chemokine ligand 2 (CCL2). Further, CCL2 recruits myeloid-derived suppressor cells to form a PMN and promote the lung metastasis of BC [83]. LncRNAs are aberrantly expressed in patients with BCs [132]. Xia W et al. have demonstrated that a high expression of EV containing lncRNA SNHG16 in BC cells and tissues promotes the EMT of BC cells via the miR-892b/PPAPDC1A axis, thereby promoting the lung metastasis of BC cells [84].

#### 3.3.2. Lung Metastasis of Digestive System Cancer

GC is the fourth leading cause of cancer-related deaths worldwide [118]. Previous studies have demonstrated the role of circRNAs in GC metastasis. CircTMEM87A acts like an oncogene and mediates its effect via the miR-142-5p/ULK1 axis in GC. The in vivo knockdown of circTMEM87A inhibits GC tumorigenicity and lung metastasis [85]. A recent study showed the high expression of exosomal circFCHO2 derived from GC cells in the serum of patients with GC. CircFCHO2 sponges miR-194-5p, thereby activating the JAK1/STAT3 pathway and promoting cell growth and lung metastasis of GC [86]. Further, exosomal circNRIP1 promotes lung metastasis in GC via the miR-149-5p/AKT1 axis [53].

CRC is the third most commonly diagnosed cancer and the second leading cause of cancer-related mortality [133]. The lung is one of the most common sites for CRC organ-specific metastasis [134]. CRC-derived exosomal miR-25-3p promotes liver and lung metastasis [76]. A study showed a significant increase in the expression of exosomal miR-106b-3p in patients with metastatic CRC. Exosomal miR-106b-3p targets genes deleted in liver cancer-1 (DLC-1), thereby promoting EMT and lung metastasis [87].

Cholangiocarcinoma (CCA) is the most common biliary malignancy, with a 50–70% mortality and recurrence rate [135]. A study by Ni Q et al. revealed an overexpression in miR-23a-3p levels in CCA cells and tissues. CCA-derived exosomal miR-23a-3p may promote tumor growth and metastasis by negatively regulating Dynamin3 [88].

#### 3.3.3. Lung Metastasis in Other Cancers

Hepatocellular carcinoma (HCC) is the third most common cause of cancer-related death worldwide [118]. The lungs are also the common metastatic site in patients with HCC. Lung metastasis indicates a rapid decline in the condition of the patient with GC [136]. Fang et al. reported that exosomal miR-103 secreted by HCC cells can be delivered to endothelial cells, which promote lung metastasis by targeting ZO-1, VE-Cadherin, and p120. This weakens the junctional integrity of endothelial cells and increases vascular permeability [89]. The HCC-derived exosomal miR-1247-3p activates the β1-integrin-NF-κB signaling pathway in fibroblasts by directly targeting B4GALT3. This activated the cancer-associated fibroblasts to secrete pro-inflammatory cytokines such as IL-6 and IL-8, which promote the lung metastasis of HCC [90].

Osteosarcoma (OS) is the most common primary malignant sarcoma of bone [137]. The lungs are the most common site for the metastasis of OS as well. Patients with OC and lung metastasis have a 5-year survival rate of approximately 20% [138]. Zhang et al. have demonstrated that miR-101 inhibits the invasion and metastasis of OS. Further, a low expression of EV-miR-101 was observed in both patients with OS and metastatic OS. EV-miR-101 directly targets BCL6, thereby inhibiting lung metastasis of OS [91]. A recent study in mice injected with BMDC-secreted EVs showed that the EV-carrying NORAD participates in the lung metastasis of OS by regulating the miR-30c-5p/KLF10 axis [92].

Cervical cancer (CC) typically occurs in the cervical epithelium. It is the second leading cause of cancer-related death in women between the age of 20–39 [139,140]. A recent study showed that EV-carrying miR-146a-5p promotes the lung metastasis of CC cells by suppressing WWC2 expression, inhibiting the phosphorylation of cofilin, and changing the depolymerization of F-actin/G-actin, thereby activating the Hippo–Yap signaling pathway [93]. Certain studies have shown that circRNAs can act as an oncogene and tumor suppressor for the progression and metastasis of CC [141,142]. Exosome-delivered circRNA_PVT1 induces EMT by targeting miR-1286, thereby promoting the invasive and migratory potential of CC cells and lung metastasis [94].

Salivary adenoid cystic carcinoma (SACC) occurs in the salivary glands [143], and lung metastasis affects the long-term survival rate of patients with SACC [144]. A study has shown a high expression of lncRNA MRPL23 antisense RNA1 (MRPL23-AS1) in patients with SACC. Moreover, exosomes carrying MRPL23-AS1 act on the zeste homolog enhancer 2 (EZH2) to form RNA-protein complexes. This leads to the inactivation of E-cadherin, which promotes microvascular permeability and EMT, thereby enhancing lung metastasis in SACC [95].

Nasopharyngeal carcinoma (NPC) occurs less frequently compared to other cancers [145]. Distant metastasis is the leading cause of mortality in patients with NPC [146]. A recent study showed that NPC cell-derived exosomal miR-205-5p targets DSC2 to enhance the expression of matrix metalloproteases (MMP), thereby promoting angiogenesis and lung metastasis in NPC cells [96].

### 3.4. Brain Metastasis

Brain metastasis (BM) most commonly occurs in lung cancers, BCs, and CRCs [147], and the prognosis of patients with BM is poor [148]. The metastatic cells should first cross the blood–brain barrier (BBB) to invade the brain parenchyma [149]. Tumor-derived EVs breach the intact BBB, which plays an important role in the progression of BM [150]. We next summarize the emerging roles and underlying mechanisms of EV-derived ncRNAs in BM (Figure 2D and Table 1).

#### 3.4.1. Brain Metastasis of Breast Cancers

BM commonly occurs in patients with BC and is associated with poor survival in patients with BC [48]. However, the mechanisms underlying brain metastasis in breast cancer (BCBM) are still unclear. Sirkisoon et al. have demonstrated that EVs secreted by BC cells can activate astrocytes. A high expression of EV-derived miR-1290 was observed in BC, which activates astrocytes in the metastatic brain microenvironment. Mechanistically, EV-derived miR-1290 inhibits the expression of transcriptional repressor FOXA2, which enhances the secretion of CNTF cytokines and astrocyte activation, thereby promoting the progression of BM [97]. Further, an increase in the expression of EVs containing miR-181c was observed in BC patients with BM. EV-containing miR-181c decreases PDPK1 expression and inhibits cofilin to promote actin degradation. This leads to the destruction of BBB and promotes BM of BC [98]. Increased BC-derived EV miR-122 expression reduces glucose uptake by downregulating the glycolysis enzyme pyruvate kinase, thereby altering the PMN and promoting brain metastasis [99]. A study has shown that BC-derived exosomal miR-105 can promote both BM and lung metastasis [81]. A high expression of lncRNA GS1-600G8.5 in exosomes derived from BC cells targets tight junction proteins to destroy the BBB and promote BM [100]. A study showed that the downregulation of XIST activates EMT and c-MET, which stimulates the secretion of exosomal miR-503. This promotes the formation of a PMN prior to BC cells metastasizing to the brain [101].

#### 3.4.2. Brain Metastasis of Lung Cancers

BM occurs in approximately 50% of patients with lung cancer, and the incidences of BM in patients with lung cancers are continuously increasing [151]. A study by Wei L et al. showed that a significant overexpression of miR-550a-3-5p in exosomes derived from lung cancers targets YAP1 to promote BM [102]. BM is a common site for metastasis in patients with NSCLC and is the primary cause of poor prognosis and the quality of life of patients with NSCLC [152]. Wu et al. revealed a significant overexpression of lnc-MMP2-2 in EVs derived from NSCLC cells. This exosomal lnc-MMP2-2 regulates TGF-β1, which promotes EndoMT, downregulates tight junction proteins, and mediates the destruction of BBB integrity. Mechanistically, exosomal lnc-MMP2-2 promotes BM in lung cancers via the miR-1207-5p/EPB41L5 axis [103].

### 3.5. Lymph Node Metastasis

Lymph node metastasis (LNM) is a well-established indicator of poor prognosis in patients with cancer, which plays a decisive role in the choice of treatment [153]. Cancer cells metastasize to sentinel lymph nodes via lymphatic vessels, thus, promoting distant metastasis [154]. Lymphatic endothelial cells regulate lymphangiogenesis and the reconstruction of the lymphatic network by the uptake of EV-containing ncRNA and other bioactive components, thereby promoting LNM [155]. In the next section, we have summarized the novel roles and underlying mechanisms of EV-derived ncRNAs in LNM (Figure 2E and Table 1).

#### 3.5.1. Lymph Node Metastasis of Bladder Cancer

Bladder cancer (BCa) is one of the most common malignancies of the genitourinary system [118]. The lymph node is the primary site for the metastasis of BCa, and the prognosis of BCa patients with LNM is extremely poor [156]. Zheng et al. have shown a significant increase in lncRNA BCYRN1 in exosomes secreted by BCa cells. Exosomal-lncRNA BCYRN1 directly binds to hnRNPA1, upregulates WNT5A expression, and activates the Wnt/β-catenin signaling pathway to promote VEGF-C secretion. Moreover, exosomal BCYRN1 transmitted to human lymphatic endothelial cells (HLEC) enhances VEGFR3 expression, which leads to the formation of the hnRNPA1/WNT5A/VEGFR3 feedforward loop. This strengthens the VEGF-C-dependent lymphangiogenesis and induces the LNM of BCa [104]. Additionally, a high expression of exosome-LNMAT2 in patients with BCa was associated with LNM in a VEGF-C-independent manner. LNMAT2 directly interacts with hnRNPA2B1 and upregulates PROX1 expression in HLEC in a VEGF-C-independent manner, thereby promoting lymphangiogenesis and LNM [105]. Furthermore, Chen et al. showed that the overexpression of exosomal ELNAT1 secreted by BCa cells could promote lymphangiogenesis and LNM. Mechanistically, ELNAT1 directly interacts with hnRNPA1 and induces UBC9 overexpression to promote the SUMOylation of hnRNPA1, thereby facilitating the packaging of ELNAT1 into EVs. Subsequently, EV-mediated ELNAT1 enhances SOX18 expression in HLECs, thereby promoting the LNM of BCa [106]. Additionally, a high expression of exosomal circPRMT5 in serum and urine of patients with urothelial carcinoma induces EMT and promotes LNM via the miR-30c/SNAIL1/E-cadherin axis [107].

#### 3.5.2. Lymph Node Metastasis of Lung Cancer

NSCLCs account for approximately 85% of all lung cancer cases, and LUAD is the most common type of NSCLC [157]. Cell migration and invasion are important characteristics of LUAD, which is the underlying cause of the high mortality rate in patients [158]. LNM is the primary cause of the poor prognosis of patients with LUAD [159]. LNM affects the treatment and prognosis of patients with NSCLC [160]. Zhou H et al. have shown the overexpression of exosomal circ RAPGEF5 in patients with LUAD, which promotes cell proliferation and LNM via the miR-1236-3p/ZEB1 axis [108]. Further, a significant correlation was observed between a high expression of HOTAIR in exosomes derived from NSCLC cells and LNM as well as a TNM stage in patients with NSCLCs [109].

#### 3.5.3. Lymph Node Metastasis of Other Cancers

LNM is an important factor that affects the prognosis and treatment of patients with cervical squamous cell carcinoma (CSCC), and CSCC is one of the most common cancers in females [161,162]. Zhou et al. demonstrated that exosomal miR-221-3p promotes lymphangiogenesis and LNM in CSCC by transporting exosomal miR-221-3p to HLECs. This is mediated by a decreased VASH1 expression, which activates the ERK/AKT signaling pathway in a VEGF-C-independent manner [110].

LNM is the primary cause of the poor prognosis of patients with esophageal squamous cell carcinoma (ESCC). The 5-year survival rate of ESCC patients with LNM reduced from 70% to 18% [163]. Liu et al. have shown that reported ESCC-derived exosomes enriched with miR-320b promote LNM by directly targeting PDCD4. This activates the AKT signaling pathway in HLECs in a VEGF-C-independent manner [111]. LNM is an important parameter for predicting the prognosis of patients with endometrial cancer (EC). Various studies have shown that an increase in the density of peritumoural lymphatic vessels correlates with metastasis and poor outcomes in patients [140,164,165]. Wang et al. has shown that a low expression of exosomal miR-26a-5p derived from EC cells was observed in the plasma of patients with EC. The uptake of exosomal miR-26a-5p by HLECs induces the formation of lymphatic vessels and promotes LNM via the LEF1/c-MYC/VEGFA axis in patients with EC [112].

## 4. Potential Clinical Applications of EV-ncRNAs in Organ-Specific Metastasis

EVs can be found in various body fluids, including blood, urine, ascites, breast milk, bile, synovial, ascites, lacrimal, seminal, and bronchoalveolar lavage [166]. A liquid biopsy is used to collect and test tumor samples using body fluids. This relatively non-invasive procedure can be used to understand primary tumors and metastases [167]. Exosomal ncRNAs extracted from body fluids of patients with cancer could serve as novel diagnostic and prognostic biomarkers as well as therapeutic targets [43,168]. The metastasis of cancer primarily contributes to the poor prognosis of patients, and exosomal ncRNAs could be used as biomarkers to predict and evaluate tumor metastasis. This could aid the managing of cancer progression in patients [169,170]. A recent study has shown that the levels of exosomal miR-193a-3p, miR-210-3p, and miR-5100 in the plasma could be used as biomarkers for distinguishing patients with metastatic lung cancer from patients with non-metastatic lung cancer. The area under curve (AUC) value was 0.8717 in receiver operating characteristic curve (ROC) analysis, indicating good specificity and sensitivity [170]. In addition, clinical data have shown that tumor exosomal integrins can determine and predict organ-specific metastasis; therefore, targeting exosome integrins could effectively inhibit organ-specific metastasis [49]. Herein, we systematically summarized EV-derived ncRNAs that could serve as potential diagnostic and therapeutic biomarkers for cancer metastasis.

### 4.1. Potential Diagnostic and Prognostic Biomarkers

Exosomes derived from cancer cells can promote metastasis by serving as carriers to transport some miRNAs from the primary site to a distant location [171]. Various studies have shown that exosomal ncRNAs derived from the body fluids of patients with LNM could serve as potential diagnostic and prognostic biomarkers. For example, a study indicated that exosomal miR-146b-5p and miR-222-3p could be used as potential biomarkers for thyroid carcinoma (PTC) patients with LNM. The results revealed a significant increase in plasma EV miR-146b-5p and miR-222-3p levels in PTC patients with LNM, and the AUC values were 0.811 and 0.834, respectively. When both miRNAs were used together in PTC screening, the AUC value increased to 0.895 [172]. A low expression of exosomal miR-363-5p was observed in BC patients with LNM, thereby downregulating PDGFB expression to inhibit LNM. Hence, miR-363-5p could be used as a potential biomarker for LNM and prognosis of patients with BC, and the AUC value was 0.733–0.958, using multiple independent datasets [173]. Further, another study indicated that exosomal miR-423-5p could be used as a potential marker for the diagnosis and prognosis of patients for GC with LNM. ROC analysis confirmed that the diagnostic efficiency of serum exosomal miR-423-5p was higher than those of CEA and CA-199. The analysis of clinicopathological parameters revealed a significant correlation between exosomal miR-423-5p levels in the serum of patients with LNM [174]. A study on 108 patients revealed that exosomal miR-10b-5p, miR-101-3p, and miR-143-5p could be used as diagnostic biomarkers for patients with GC and LNM, GC with ovarian metastasis, and GC with liver metastasis with an AUC value of 0.8247–0.8919 [175].

In addition to small RNAs, large ncRNAs also serve as potential diagnostic and prognostic biomarkers to predict and evaluate tumor metastasis. For example, a study revealed that plasma exosomal lncRNA HOTTIP levels were significantly high in patients with GC. Further, a positive correlation was observed between exosomal lncRNA HOTTIP levels and TNM staging, with an AUC value of 0.827. Therefore, plasma exosomal lncRNA HOTTIP could serve as a potential biomarker for the diagnosis and prognosis of patients with GC [176]. To evaluate the potential diagnostic value of EV-circRNAs, He et al. collected 41 LUAD tissues from 21 patients with LNM and 20 patients without LNM. The results revealed that EV-circRNA-0056616 in the plasma could be used as a potential biomarker for predicting LNM in patients with LUAD, with an AUC value of 0.812 [177].

### 4.2. Potential Clinical Value of EV-ncRNAs in Anti-Metastasis Therapy

Recently, the use of EVs as vehicles for the targeted delivery of therapeutic molecules such as small interfering RNAs (siRNAs), small compounds, and proteins is being actively explored [178,179]. Exosomal miRNAs could be used as promising noninvasive molecules for targeted cancer therapy [180]. The exosomes loaded with ncRNAs as natural nanocarriers have organic properties; hence, exosomal-ncRNAs have attracted attention for their use in anti-cancer organotropic metastasis therapeutics.

Recent studies have shown that targeting exosomal-ncRNAs can inhibit organ-specific metastasis. For example, a high expression of exosomal miR-122 secreted by patients with BC can inhibit glucose uptake by normal cells in the PMN. The in vivo inhibition of miR-122 can restore glucose uptake by distant organs, including the brain and lungs, and inhibit metastasis [99]. In a nude mouse model of lung metastasis, BC cell secrete EV-packaged miR-126 that can be administered intravenously to produce a lung homing effect, inhibiting lung metastasis by disrupting the PTEN/PI3K/AKT signaling pathway [181]. A recent study showed that anti-miR-105 treatment could reduce the size of the primary BC, and inhibit brain and lung metastasis [81]. A study revealed that plasma EV-miR-101 serves as a promising biomarker for predicting OS metastasis (AUC = 0.8307). Further, in a nude mice xenograft injected with adipose tissue-derived MSCs, EV-miR-101 could serve as a novel therapeutic strategy for patients with lung metastasis of OS [91]. Zeng et al. revealed that exosomal circ FNDC3B inhibits tumor growth and liver metastasis of CRC by regulating miR-937-5p/TIMP3 axis. Furthermore, a decrease in tumor volume, weight, and the number of metastatic nodules in the liver were observed in mice injected with exosomal circFNDC3B, thereby inhibiting CRC liver metastasis [182]. Hence, these engineered exosomes loaded with oncogenic or tumor-suppressing ncRNAs could be used as strategies for the targeted treatment of metastatic organotropism.

## 5. Conclusions and Perspectives

Our review highlights the roles and underlying mechanisms of EV-derived ncRNAs in organ-specific metastasis. Further, these EV-derived ncRNAs could serve as potential diagnostic and prognostic biomarkers and could be used in the treatment of cancer metastasis. Studies have shown that EV-derived ncRNAs can alter organotropic metastasis and serve as promising non-invasive biomarkers for the diagnosis and prognosis of cancers. Furthermore, targeting EV-ncRNAs can inhibit metastases in various cancers, which indicates that EV-ncRNAs could be used in anti-cancer therapeutics.

However, a series of key questions are yet to be addressed: (i) Studies have proved that EV-ncRNAs could serve as promising non-invasive biomarkers for the diagnosis and prognosis of metastatic cancer. However, a standardized protocol for EV isolation is challenging and should be addressed for its successful implementation in clinical settings. Further, the current exosomal isolation and purification techniques are inadequate. Traditionally, exosome isolation techniques include ultra-high speed centrifugation, ultrafiltration, exosome precipitation, and immune-enrichment [24]. High-tech equipment and materials required for this complex and time-consuming procedures. Moreover, low productivity, and purity are some of the disadvantages associated with these techniques. Therefore, more advanced exosome extraction techniques are needed to accelerate the research and potential application of exosomal ncRNAs. Furthermore, techniques such as microfluidic devices, nano-plasma-enhanced scattering, membrane-mediated exosome isolation, and lab-on-a-chip devices are potential approaches to accelerate translational research of EVs in cancer diagnosis and treatment. (ii) Studies have shown that exosomal ncRNAs derived from tumor cells are usually significantly upregulated, and the inhibition or knockdown of exosomal ncRNAs can alter tumor progression and organotropic metastasis. Therefore, anti-cancer therapy targeting exosomes or exosomal ncRNAs is feasible. However, the studies on exosome-ncRNA are currently focusing on bioinformatics analysis, sample sequencing and testing, and preclinical studies such as in vitro and in vivo experiments [183]. Therefore, prospective, large-scale, multi-center, carefully designed clinical trials should be conducted. (iii) EVs can be processed as carriers for the targeted delivery of potential therapeutic agents. However, EVs can promote and inhibit tumor growth and metastasis by transferring ncRNAs; hence, the use of EVs in therapeutics can be challenging. Therefore, further studies are required to evaluate the packaging effect and specific uptake of non-toxic EVs such as red blood cell derived EVs [184]. 

EV-ncRNAs can serve as promising biomarkers for predicting and diagnosing organ-specific metastasis. Additionally, they can be used as a targeted therapeutic agent or vehicle for gene delivery in organ-specific metastasis therapeutics. Further studies on EV-ncRNAs, which can aid in improving the survival rate and prognosis of patients with metastatic cancer, are required.

## Figures and Tables

**Figure 1 cancers-14-05693-f001:**
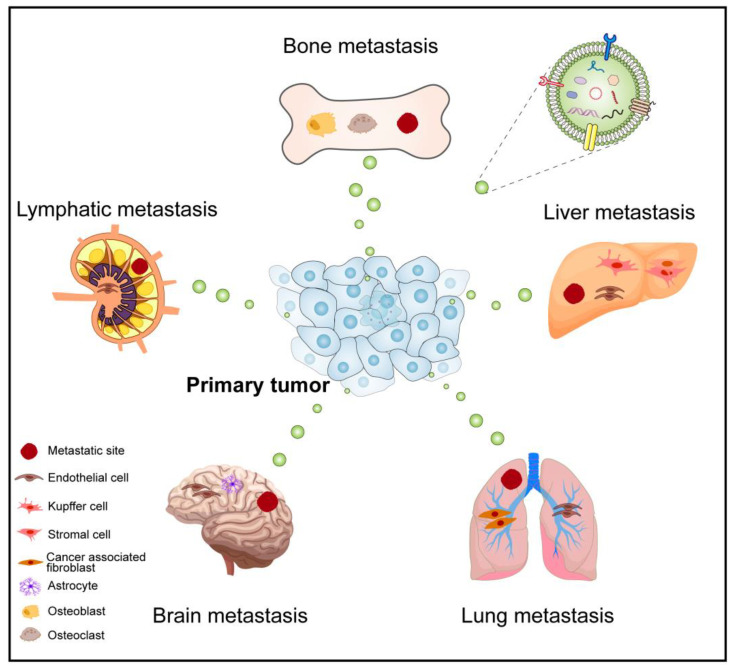
Tumor cell-derived EVs mediate organ-specific metastasis. Primary tumor cells secrete EVs into circulation as messengers, which spread and colonize to different metastatic sites, including bone, liver, lung, brain, and lymph nodes.

**Figure 2 cancers-14-05693-f002:**
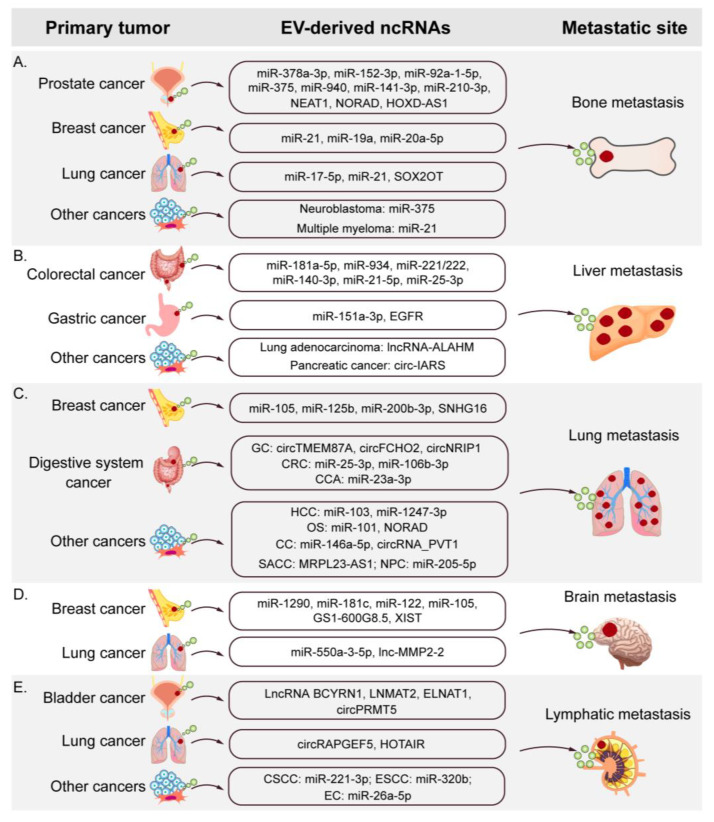
Tumor cell-derived EVs transfer different ncRNAs to induce organ-specific metastasis. Primary tumor cell-derived EVs deliver ncRNAs to specific organs or sites, which mediate organ-specific metastasis, including bone metastasis (**A**), liver metastasis (**B**), lung metastasis (**C**), brain metastasis (**D**), and lymphatic metastasis (**E**).

**Table 1 cancers-14-05693-t001:** The emerging roles of EV-derived ncRNAs in organ-specific metastasis.

Primary Tumor	EV-ncRNA	Expression	Molecular Axis	Functions	References
Bone metastasis					
Prostate cancer	miR-378a-3p	Up	Activates the Dyrk1a/Nfatc1/Angptl2 axis	Promotes osteolytic progression	[54]
	miR-152-3p	Up	Targets osteoclastogenic regulator MAFB	Promotes osteolytic progression	[55]
	miR-92a-1-5p	Up	Targets COL1A1	Promotes osteoclast differentiation and suppresses osteoblastogenesis	[56]
	miR-375	Up	-	Stimulates the activity of osteoblast	[57]
	miR-940	Up	Targets ARHGAP1 and FAM134A	Induces osteoblast phenotypic differentiation	[58]
	miR-141-3p	Up	Targets DLC-1, activates the p38MAPK pathway.	Promotes the activity of osteoblasts and inhibits osteoclasts	[59]
	miR-210-3p	Up	Targets negative regulators of NF-κB signaling TNIP1 and SOCS1	Promotes bone metastasis	[60]
	NEAT1	Up	miR-205-5p/RUNX2 and SFPQ/PTBP2 axis	Promotes osteogenic differentiation	[61]
	NORAD	Up	the miR-541-3p/PKM2 axis	Promotes bone metastasis	[62]
	HOXD-AS1	Up	the miR-361-5p/FOXM1 axis	Promotes bone metastasis	[52]
Breast cancer	miR-21	Up	Targets PDCD4	Regulates the generation of osteoclasts and establishes the PMN	[63]
	miR-19a	Up	Assists with IBSP	Induces osteoclast generation and creates a microenvironment favorable for bone metastasis	[64]
	miR-20a-5p	Up	Targets SRCIN1	Promoted osteoclast generation	[65]
Lung cancer	miR-17-5p	Up	Targets PTEN, actives PI3K/Akt pathway	Promotes osteoclastogenesis and bone metastasis	[66]
	miR-21	Up	Targets PDCD4	Promotes osteoclastogenesis	[67]
	SOX2OT	Up	Targets miR-194-5p/RAC1 signaling axis and regulates TGF-β/pTHrP/RANKL pathway	Modulates osteoclast differentiation	[68]
Neuroblastoma	miR-375	Up	Targets YAP1	Promotes osteogenic differentiation	[69]
Multiple myeloma	miR-21	Up	Activates STAT3 (PIAS3)	Induces osteoclastogenesis	[70]
Liver metastasis					
Colorectal cancer	miR-181a-5p	Up	Targets SOCS3 and activates the IL6/STAT3 signaling pathway	Activates hepatic stellate cells, remodels the TME and promotes liver metastasis	[71]
	miR-934	Up	Downregulates PTEN expression and activates the PI3K/AKT signaling pathway	Promotes PMN formation and promotes liver metastasis	[72]
	miR-221/222	Up	Activates liver HGF by suppressing SPINT1 expression	Promotes PMN formation and promotes liver metastasis	[73]
	miR-140-3p	Down	Targets BCL9 and BCL2	Suppress liver metastasis	[74]
	miR-21-5p	Up	the miR-21-TLR7-IL-6 axis.	Induces an inflammatory PMN, promotes liver metastasis	[75]
	miR-25-3p	Up	Targets KLF2 and KLF4, increases the expression of VEGFR2 and decreases the levels of ZO-1, OCCLUDIN, and CLAUDIN 5	Promotes vascular permeability and angiogenesis, induces PMN formation and liver metastasis	[76]
Gastric cancer	miR-151a-3p	Up	Targets YTHDF3, activates the SMAD2/3 pathway	Promotes PMN formation and liver metastasis	[77]
	EGFR	Up	Downregulates the expression of miR-26a/b, activates HGF	Regulates the liver microenvironment and promotes liver metastasis	[78]
Lung adenocarcinoma	lncRNA- ALAHM	Up	Binds with AUF1	Promotes hepatocyte secretion of HGF and liver metastasis	[79]
Pancreatic cancer	Circ-IARS	Up	Downregulates miR-122 and ZO-1 levels, increases Rho A activity and F-actin expression	Regulate endothelial monolayer permeability to promote liver metastasis	[80]
Lung metastasis					
Breast cancer	miR-105	Up	Inhibits tight junction protein ZO-1	Induces vascular permeability and promotes lung metastasis	[81]
	miR-125b	Up	Suppresses TP53INP1 and TP53	Activates cancer-associated fibroblasts	[82]
	miR-200b-3p	Up	Targets PTEN activates the AKT/NF-κBp65 pathway	Promotes BC PMN and lung metastasis	[83]
	SNHG16	Up	the miR-892b/PPAPDC1A axis	Promotes EMT and lung metastasis	[84]
Gastric cancer	circTMEM87A	Up	the miR-142-5p/ULK1 axis	Promotes lung metastasis	[85]
	circFCHO2	Up	Activates the miR-194-5p/JAK1/STAT3 pathway	Promotes lung metastasis	[86]
	circNRIP1	Up	the miR-149-5p/AKT1/mTOR axis	Promotes lung metastasis	[53]
Colorectal cancer	miR-25-3p	Up	Targets KLF2 and KLF4	Induces PMN formation and promotes lung metastasis	[76]
	miR-106b-3p	Up	Inhibits DLC-1	promotes EMT and lung metastasis	[87]
Cholangiocarcinoma	miR-23a-3p	Up	Inhibits Dynamin3	Promotes lung metastasis	[88]
Hepatocellular carcinoma	miR-103	Up	Targets ZO-1, VE-Cadherin and p120	Increases vascular permeability and promotes lung metastasis	[89]
	miR-1247-3p	Up	Targets B4GALT3, activates the β1-integrin-NF-κB signaling pathway	Promotes lung metastasis	[90]
Osteosarcoma	miR-101	Down	Inhibits BCL6	Suppresses lung metastasis.	[91]
	NORAD	Up	Regulates the miR-30c-5p/KLF10 axis	Promotes lung metastasis.	[92]
Cervical cancer	miR-146a-5p	Up	Inhibits WWC2 to activate the Hippo–YAP pathway	Promotes lung metastasis	[93]
	circRNA_PVT1	Up	Inhibits miR-1286	Induces EMT and promotes lung metastasis	[94]
Salivary adenoid cystic carcinoma	MRPL23-AS1	Up	Forms an RNA-protein complex with EZH2	Enhances microvascular permeability, promotes EMT and lung metastasis	[95]
Nasopharyngeal carcinoma	miR-205-5p	Up	Targets DSC2 to enhance the EGFR/ERK signaling and MMP2/MMP9 expression	Promotes angiogenesis and lung metastasis	[96]
Brain metastasis					
Breast cancer	miR-1290	Up	the FOXA2/CNTF axis	Activates astrocytes in the brain metastatic microenvironment, promote brain metastasis	[97]
	miR-181c	Up	Targets PDPK1 and inhibits cofilin	Destroies BBB integrity, promotes brain metastasis	[98]
	miR-122	Up	Inhibits the glycolysis enzyme pyruvate kinase	Promotes the PMN and brain metastasis	[99]
	miR-105	Up	Inhibits tight junction protein ZO-1	Induces vascular permeability and promotes brain metastasis	[81]
	GS1-600G8.5	Up	Targets tight junction proteins	Destroies the BBB system and promotes brain metastasis	[100]
	XIST	Down	miR-503, activates MSN-c-Met	Promotes a PMN formation and brain metastasis	[101]
Non-small cell lung cancer	miR-550a-3-5p	Up	Inhibits YAP1	Promotes brain metastasis	[102]
	lnc-MMP2-2	Up	the miR-1207-5p/EPB41L5 axis	Destroies BBB integrity, promotes brain metastasis.	[103]
Lymph node metastasis					
Bladder cancer	LncRNA- BCYRN1	Up	hnRNPA1/WNT5A/VEGFR3	Promotes VEGF-C-dependent lymphangiogenesis and lymphatic metastasis	[104]
	LNMAT2	Up	Recruits hnRNPA2B1, upregulates PROX1 expression	Promotes VEGF-C-independent lymphangiogenesis and lymphatic metastasis	[105]
	ELNAT1	Up	Interacts with hnRNPA1, activate the ELNAT1/UBC9/SOX18 regulatory axis	Promotes lymphangiogenesis and lymph node metastasis	[106]
	circPRMT5	Up	the miR-30c/SNAIL1/E-cadherin pathway	Induces EMT, promotes lymphatic metastasis	[107]
lung cancer	circRAPGEF5	UP	the miR-1236-3p/ZEB1 axis	Promotes lymph node metastasis	[108]
	HOTAIR	Up	-	Promotes lymphatic metastasis	[109]
Cervical squamous cell carcinoma	miR-221-3p	Up	the miR-221-3p-VASH1 axis, the ERK/AKT pathway	Promotes lymphangiogenesis and lymphatic metastasis	[110]
Esophageal squamous cell carcinoma	miR-320b	Up	Targets PDCD4, activates the AKT pathway	Promotes VEGF-C- independent lymphangiogenesis and lymphatic metastasis	[111]
Endometrial cancer	miR-26a-5p	Down	LEF1/c-MYC/VEGFA axis	Induces lymphatic vessel formation, promotes lymphatic metastasis	[112]

PDCD4, programmed cell death 4; PMN, pre-metastatic niche; TME, tumor microenvironment; HGF, hepatocyte growth factor EGFR; BCL2, B-cell lymphoma-2; ZO-1, zonula occludens-1; EGFR, epidermal growth factor receptor; EMT, epithelial-mesenchymal transition; DLC-1, deleted in liver cancer-1; MRPL23-AS1, MRPL23 antisense RNA1; EZH2, zeste homologue enhancer 2; MMP, matrix metalloproteases; BBB, blood-brain barrier.

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
