# Peer review of "Non-Coding RNAs of Extracellular Vesicles: Key Players in Organ-Specific Metastasis and Clinical Implications"

_cancers, 2022, doi:10.3390/cancers14225693_

Round 1

Reviewer 1 Report

In this review, the authors highlight that EVs-ncRNAs play a key role on organ-specific metastasis and clinical implications. This is an interesting topic with clinical significance. Firstly, the classification and biological characteristics of EVs were introduced. The authors emphasizes the important role of ncRNAs in function of EVs. Then the mechanism of EVs-ncRNAs secreted by different tumors enhancing organ-specific metastases of liver, bone, lung, brain, and lymph nodes was elucidated. Moreover, the authors summarize clinical application of EVs-ncRNAs served as potential biomarkers for diagnosis, prognosis and therapy.

EVs-ncRNAs is a novel direction in the field of cancer research, which has great application prospects in the field of cancer diagnosis and treatment. The manuscript is well organized and the English writing is clear. However, the manuscript requires minor revisions on English writing.

1. Please define the abbreviations in the text when used for the first time. For example, In page 9 ,line 244 “NSCLC”.

2. I suggest that all the tumor-associated EV-ncRNAs summarized in Table 1 need be presented in Figure 2. For example, miR-103 from breast cancer EVs.

Author Response

In this review, the authors highlight that EVs-ncRNAs play a key role in organ-specific metastasis and clinical implications. This is an interesting topic with clinical significance. Firstly, the classification and biological characteristics of EVs were introduced. The authors emphasize the important role of ncRNAs in the function of EVs. Then the mechanism of EVs-ncRNAs secreted by different tumors enhancing organ-specific metastases of liver, bone, lung, brain, and lymph nodes were elucidated. Moreover, the authors summarize the clinical application of EVs-ncRNAs served as potential biomarkers for diagnosis, prognosis, and therapy.

EVs-ncRNAs are a novel direction in the field of cancer research, which has great application prospects in the field of cancer diagnosis and treatment. The manuscript is well organized and the English writing is clear. However, the manuscript requires minor revisions in English writing.

Point 1: Please define the abbreviations in the text when used for the first time. For example, On page 9, line 244 “NSCLC”.

Response: We thank the reviewer for the expert insights. We have defined the abbreviations of “NSCLC” on page 9, line 252. Meanwhile, we have checked the abbreviations in the whole text to make sure that there are no errors.  

Point 2: I suggest that all the tumor-associated EV-ncRNAs summarized in Table 1 need to be presented in Figure 2. for example, miR-103 from breast cancer EVs.

Response: Thanks to the experts for the questions, and apologies for our mistakes. We have ensured all the tumor-associated EV-ncRNAs are correctly summarized in Table 1 and presented in Figure 2. In addition, exosomal miR-122 in brain metastasis of breast cancers has been added in Table 1.

Reviewer 2 Report

The manuscript of Qian Jiang and Xiao-Ping Tan et al. summarized the function of non-cording RNAs contained in extracellular vesicles in cancer metastasis and their therapeutic and diagnostic potential. A wide range of recent reports have been summarized well in an easy-to understand manner, but I suggest some minor concerns.   

Minor concerns

1.  In Figure 1, endothelial cells need to be added to liver metastasis according to reference 76 and 80. In addition, the explanation for the red circle description in lung metastasis is missing.

ï¼’.  In Table 1, please add an explanation to the legend about the abbreviation of PMN.

Author Response

The manuscript of Qian Jiang and Xiao-Ping Tan et al. summarized the function of non-coding RNAs contained in extracellular vesicles in cancer metastasis and their therapeutic and diagnostic potential. A wide range of recent reports have been summarized well in an easy-to-understand manner, but I suggest some minor concerns.

Minor concerns

Point 1: In Figure 1, endothelial cells need to be added to liver metastasis according to references 76 and 80. In addition, the explanation for the red circle description in lung metastasis is missing.

Response: We thank the reviewer for the expert insights. Apologies for our unclear presentation. In the revised manuscript of Figure 1, we have added endothelial cells to liver metastasis and removed the red circle in lung metastasis.

Point 2: In Table 1, please add an explanation to the legend about the abbreviation of PMN.

Response: Thanks to the experts for the questions, and apologies for our mistakes. In Table 1, we have added an explanation to the legend about the abbreviation of PMN. In addition, we carefully checked the abbreviations of PMN in the revised manuscript.